# A Review on the Physical Parameters Affecting the Bond Behavior of FRP Bars Embedded in Concrete

**DOI:** 10.3390/polym14091796

**Published:** 2022-04-28

**Authors:** Boğaçhan Başaran, İlker Kalkan, Ahmet Beycioğlu, Izabela Kasprzyk

**Affiliations:** 1Department of Construction, Vocational School of Technical Sciences, Amasya University, Amasya 05100, Turkey; bogachan.basaran@amasya.edu.tr; 2Department of Civil Engineering, Faculty of Engineering and Architecture, Kırıkkale University, Kirikkale 71450, Turkey; 3Department of Civil Engineering, Faculty of Engineering, Adana Alparslan Türkeş Science and Technology University, Adana 01250, Turkey; abeycioglu@atu.edu.tr; 4Faculty of Civil and Environmental Engineering and Architecture, Bydgoszcz University of Science and Technology, Al. Prof. S. Kaliskiego 7, 85-796 Bydgoszcz, Poland; izabela.kasprzyk@pbs.edu.pl

**Keywords:** mechanical interlocking, ribbed surface, surface friction, polymer reinforcement, thermal expansion, wound bar, epoxy resin, bond behavior

## Abstract

The present study is a detailed literal survey on the bond behavior of FRP (Fiber Reinforced Polymer) reinforcing bars embedded in concrete. There is an urgent need for the accurate assessment of the parameters affecting the FRP–concrete bond and quantification of these effects. A significant majority of the previous studies could not derive precise and comprehensive conclusions on the effects of each of these parameters. The present study aimed at listing all of the physical parameters affecting the concrete-FRP bond, presenting the effects of each of these parameters based on the common opinions of the previous researchers and giving reasonable justifications on these effects. The studies on each of the parameters are presented in detailed tables. Among all listed parameters, the surface texture was established to have the most pronounced effect on the FRP–concrete bond strength. The bond strength values of the bars with coarse sand-coating exceeded the respective values of the fine sand-coated ones. However, increasing the concrete strength was found to result in a greater improvement in bond behavior of fine sand-coated bars due to the penetration of concrete particles into the fine sand-coating layer. The effects of fiber type, bar diameter and concrete compressive strength on the bar bond strength was shown to primarily originate from the relative slip of fibers inside the resin of the bar, also known as the shear lag effect.

## 1. Introduction

Fiber-reinforced polymer (FRP) bars have been increasingly used in the field of structural engineering due to their significant advantages, such as high tensile and fatigue strengths, high corrosion resistance, lightweight, ease of transportation and handling, thermal and electrical insulating (GFRP only) properties and being unresponsive to magnetic fields [1,2,3]. On the contrary, FRP reinforcing bars have certain important disadvantages, including creep failure, the scarce commercial availability, high production cost, anisotropic material properties, limited ductility, low modulus of elasticity, intolerance to bending (for the use as stirrups) and low transverse and lateral strength values as compared to the longitudinal tensile strength [4,5,6]. Additionally, the dearth or limited extent of the provisions in the existing FRP reinforced concrete (RC) codes and regulations constitutes another limitation for the structural use of FRP bars. Considering the numerous favorable effects of FRP bars on the service lives of structures, the national economy and the natural environment, researchers are striving to devise methods to overcome the shortcomings related to the structural use of FRP bars and promoting their use as concrete reinforcement.

The previous research studies and field applications concentrated on four different types of FRP (GFRP, BFRP, CFRP and AFRP) that can be used in the form of concrete reinforcement. BFRP and GFRP bars are the most preferred types as internal reinforcement in concrete members owing to their lower prices and ease of supply than the CFRP and AFRP bars. BFRP bars have slightly higher modulus of elasticity and tensile strength values than GFRP bars, yet the respective values of both BFRP and GFRP are considerably lower than those of CFRP and AFRP [3,7]. GFRP bars are highly vulnerable to corrosion, fatigue and creep [5]. In general, the long-term performances of GFRP bars under the coupled effects of environmental factors (temperature, humidity and corrosive medium) and loading is poor, meaning that the bond degradation of GFRP bars in concrete is more pronounced as compared to the other three types. AFRP bars, which are known to have better mechanical properties and long-term performances in comparison with the GFRP and BFRP bars, are highly vulnerable to UV effects [7,8,9]. Furthermore, the standards and regulations generally do not contain AFRP bars due to the dearth of research studies on these bars.

CFRP has various superiorities over the other three types, including the highest tensile and fatigue strengths and being the least vulnerable FRP type to environmental effects (humidity, corrosion and temperature), fatigue and creep rupture. Nevertheless, CFRP also has some major disadvantages, including the electric conductivity, high price, vulnerability to electrochemical corrosion when in contact with metal materials in a humid environment and the highly brittle nature [3,4]. Recently, carbon/glass hybrid FRP (HFRP) bars [10,11], which cater for the durability needs through the use of carbon fibers and reduce the overall cost through the use of glass fibers, and platelet reinforced composites [12,13] received widespread attention in the academia and practice. HFRP bars allow the utilization of the advantages of different FRP materials in the same reinforcing bar.

The bond between an FRP bar and the surrounding concrete is the governing factor that determines the efficiency and suitability of the utilization of FRP bars as concrete reinforcement. In flexural RC members (slabs and beams), the compression forces in concrete are counterbalanced by the tension forces in the reinforcement and the development of these tension forces entails the adequacy of the reinforcement–concrete bond in the tension zone. The types of bond mechanism of FRP bars in concrete are similar to those of steel bars, which are the mutual adhesion, surface friction and shear interlock. Nevertheless, the mechanical properties of FRP bars are completely different from those of steel bars (Figure 1) [14,15]. Therefore, the flexural behavior of RC members with steel reinforcement cannot serve as a basis for the evaluation of flexural behavior of FRP RC members. Characterization of the FRP–concrete bond is the prevailing factor in determining the ductility, bending capacity and energy absorption capacity values of FRP RC members [16,17]. The present study is a rather detailed summary on all of the previous studies in the literature on the key factors affecting the bond behavior of FRP bars in concrete.

## 2. Aims and Scope

In the literature, numerous experimental, analytical and numerical studies were conducted on identifying the prominent factors affecting the concrete-FRP bond; nevertheless, a great majority of these studies concentrated on monitoring the influence of only one or a few of all parameters. In the current study, on the contrary, all main parameters affecting the bond strength of FRP bars in concrete were investigated by conducting an extended literature survey. This survey indicated that there is consensus among the researchers on the effects of certain parameters on the FRP–concrete bond, while the effects of the other parameters have not been clearly unveiled yet. There is a variety of opinions on the effects of these parameters. The present literature review introduces the parameters one by one together with the findings of the previous researchers on this parameter. Besides, the related provisions from the structural FRP RC codes are also discussed throughout the manuscript. Accordingly, the main goal of the present study is to present the recent developments and challenges related to the utilization of FRP bars in concrete by underscoring the most influential factors affecting the FRP–concrete bond based on the previous works on this topic.

There are a total of 10 main parameters affecting the FRP–concrete bond. These parameters can be listed under four headings, which are (i) the inherent properties of FRP rebars; (ii) the arrangement and configuration of reinforcement; (iii) the inherent properties of concrete; and (iv) the method of testing. The complete list of parameters is as follows:Inherent properties of FRP
Bar Diameter (*d*);Fiber Type and Modulus of Elasticity;Surface texture of the rebar.Reinforcement Arrangement and Configuration
Concrete cover (*C*) and bar spacing (*s_c_*);Development (*l_d_*) or embedment length (*l_e_*);Reinforcement position in the member;The presence of transverse reinforcement.Inherent properties of concrete
Compressive strength (*f_c_*);The presence and percentage of fibers;The type of concrete.

A detailed section is devoted to each one of these parameters in the following discussion. A detailed table, which compiles the tests and studies used in that section, is given in each section to avoid any confusion and to clearly reveal the effects of each parameter on the FRP–concrete bond. The entire table of each section was discussed and debated in its entirety with additional comments of the authors.

The previous researchers did not reach an agreement on the denominations of the surface textures of FRP bars. In other words, the same surface type was termed differently by different researchers. This discrepancy caused significant confusion among researchers. To avoid confusion in the present review, each table contains two columns for the bar surface notations. The first column corresponds to the original notations in the source papers, while the second one refers to the notations suggested and used in the present text. The bar surface notations of the present review are given in Section 3.1.3 in detail for the sake of clarity.

The graphs and test results in the majority of the previous studies cannot provide credential comments on the effects of a certain test parameter on bond strength. The previous experimental studies do not possess the merit of focusing on a single parameter by isolating the related experiments from the remaining test parameters. Consequently, the experiments, which are intended to unfold the effects of a certain parameter on bond strength, include the coupled effects of numerous parameters. Moreover, the average values of the test results were adopted in the previous works when explicating the scatter plots. However, the values in these plots are scattered in a broad range and reliable and precise results may not be inferred by using the average values. Unlike the other review studies, the coupling of effects of different parameters were taken into account in the present text and the findings were elaborated by avoiding controversial arguments.

In this study, as much data as possible was compiled from the literature to prepare tables from which clear and accurate conclusions on the effects of a single parameter could be reached. In this way, the effects of the remaining parameters on the FRP–concrete bond were minimized, if not completely eliminated. Furthermore, each finding or conclusion was justified with sufficient reasoning. The authors did not utilize scatter graphs or curves. Furthermore, the authors avoided using precise statements on certain parameters due to significant discrepancies between the related experiments in the literature. The ambiguous and even sometimes opposing findings in different studies complicates to draw definitive conclusions on these parameters. These discrepancies have been completely ignored in the previous statistical review studies. The present paper leaves it to the readers on these parameters instead of deriving conclusions from the controversial data. Only obvious and well-explained conclusions were drawn according to the existing test results, presented in tables in each section. In the present study, the dependent and independent (bond strength) variables needed to be presented in the table format rather than conducting an analysis and presenting them in a mathematical form for three main reasons:The effect of a single parameter (dependent variable) on the independent variable (the FRP–concrete bond strength in this case) can only be unfolded if all other dependent variables are kept fixed in the related experiments. Otherwise, the coupling between the effects of several parameters will not allow the researchers to isolate of the effect of a single parameter and set a relationship between the examined dependent variable and the independent variable. In the context of investigating the effects of the FRP material type on the FRP–concrete bond strength, for instance, the surface texture, diameter, clear cover and distance from the adjacent bar of the tested bars need to be kept identical in the related experiments as well as the concrete grade, concrete type and fiber content of the concrete mixture. In that respect, the existing experiments in the literature do not suffice for the development of specific relations between each test parameter and the FRP–concrete bond strength.The test data on FRP–concrete bond strength is extremely scattered. The wide dispersion of this data mainly stems from the coupling between the effects of several parameters in the previous experimental studies, which were designed without paying attention to all parameters affecting the FRP–concrete bond. The mathematical analyses on the data with such a dispersion do not generate meaningful and accurate expressions, since the deviation of the actual data from the mathematical curve remains high, meaning that the mathematical curve does not accurately represent the experimental data.The surface texture types of FRP bars have not been standardized with regulations, standards and previous experimental studies. For instance, the rib dimensions of the ribbed FRP bars and the grain sizes of the coating layer in the sand-coated bars are rather different in different studies. Hence, the surface type with the same notation can be excessively different in the related tests, which exacerbates the broad scatter of the test data and even results in opposing test data in different experimental studies.

The authors could not make separate analyses on each type of FRP bars (CFRP, GFRP, BFRP and AFRP) in each section due to the absence of an adequate number of studies in the literature. For instance, the number of studies on CFRP and AFRP reinforcing bars is so limited that conducting separate analysis and reaching specific conclusions on these two types is not possible at all. Strictly speaking, the majority of the studies in the literature pertain to the bond behavior of GFRP and BFRP reinforcing bars in concrete. Consequently, the authors tried to reach general conclusions on the effects of each parameter on the FRP–concrete bond without diving into special comments on different FRP types. The findings from a single type of FRP were not generalized to all types, but only the common conclusions on all FRP types were given in the manuscript.

The present pertains solely to the short-term bonding performances of FRP reinforcing bars in concrete. The FRP–concrete interfacial bond strength is subject to degradation due to environmental factors, including but not limited to the temperature, corrosive environments and humidity. Furthermore, long-term effects, including creep and fatigue, are also responsible for the changes in the adherence of FRP bars to concrete. The long-term bonding performances of FRP bars and the durability issues are planned to be covered in a companion paper.

## 3. Physical Parameters Affecting the FRP–Concrete Bond Behavior

### 3.1. Inherent Properties of FRP Materials

#### 3.1.1. Bar Diameter

Significant effort has been spent in the literature to determine the effects of FRP bar diameter on bond strength. There are three basic opinions on the influence of bar diameter on bond strength. The first of these opinions relies on the concept of the relative slip between the core and the fibers on the outer surface (shear lag effect) resulting from the low slip resistance within the epoxy resin and at the epoxy–fiber interface under the action of axial tension forces [18,19,20,21] (Figure 2). As the second view, a greater amount of water is assumed to accumulate underneath the FRP bar with increasing bar diameter. The increasing amount of water causes the total volume of gaps in the mixture to increase, which in turn will reduce the FRP–concrete bond strength [22,23,24]. The last view is related to the Poisson’s effect. With increasing bar diameter, the Poisson’s effect increases and the larger decrease in the bar volume due to this effect is assumed to cause greater reductions in the mechanical interlocking and friction forces on the bar surface [21,25]. Table 1 presents the statistical, review and research studies on the effects of bar diameter on the FRP–concrete bond. This table clearly depicts that the degree of this effect covers a wide range. Even in studies with identical surface texture, concrete compressive strength and embedment length, significant differences were reported on the degree of influence of bar diameter on bond strength.

In this respect, the main reason for the differences between the findings of different researchers are expected to be a result of the shear lag effect. Several inherent properties of an FRP bar, including the fiber density, resin type, resin density and the mechanical properties of the constituents, were found to impinge on the shear lag effect. The degree of this effect also changes with the manufacturing conditions of the bar and the persistence of these conditions. Hence, the effects of bar diameter on bond strength can only be identified by considering all these variables.

#### 3.1.2. Fiber Type and Modulus of Elasticity

The literature contains a lot of studies on the influence of fiber type (Aramid, Basalt, Carbon and Glass) on FRP–concrete bond strength. The statistical, review and research studies on this very topic are presented in Table 2. The generality of these studies pertains to the bond behavior of FRP bars (Figure 3) and comparison of this behavior to the respective behavior of the reference deformed steel rebars. As a matter of fact, Table 2 tabulates the comparison of the bond strength values of different types of FRP bars with each other and with proper ribbed steel bars. The table showcases that the bond strength values of GFRP bars (except the smooth ones) vary in the range of 0.45–1.07, BFRP bars in the range of 0.36–1.46, and CFRP bars in the range of 0.60–1.35 times the respective strength values of the comparable ribbed steel bars. This comparison does not embrace the AFRP bars due to dearth of studies on these bars. Independent from the fiber type, the bond strength values can be observed to vary in a broad range, most probably due to the non-uniformity of the surface textures of the bars in these studies. The same comparison is also elaborated in Section 3.1.3, since the surface texture of the bar is much more influential on the bond strength than the other parameters.

In order to determine the effect of fiber properties on bond strength, Table 2 also compares the test results of the bars with identical bar diameter, surface texture and test conditions but with different fiber type. This comparison indicates that the bond strength values of CFRP bars range between 0.92 and 2.83 times the respective values of the GFRP bars, while the bond strengths of BFRP bars lie in the range of 1.29–1.88 times the strength values of their GFRP counterparts. The bond strength values of the CFRP bars, on the other hand, remain in the interval of 0.71–1.51 times those of the BFRP counterparts. Finally, the related strength values of CFRP rebars change from 1.16 to 1.69 times the bond strength values of the AFRP rebars with identical features.

The only clear outcome from this comparison is that the GFRP bars is the least favorable polymer bars in terms of adherence with concrete among the four types of FRP. However, reaching a crystal-clear conclusion about the contribution of AFRP, BFRP and CFRP fibers to the adherence with concrete is impossible to reach based on the available test results. The improved bonding properties of CFRP, BFRP and AFRP bars originate from two main reasons. The first reason is the shear lag effect, just like the related discussion on the effect of bar diameter. The shear lag effect is lower in FRP bars with resin and fibers strong in tension as compared to those with resin and fibers weak in tension. Considering that AFRP, BFRP and CFRP bars generally possess higher tensile stiffness values in comparison to the GFRP rebars [42,43], the probable shear lag in GFRP bars might have resulted in the reduced bond strength values in concrete. Secondly, the transfer of internal forces from the surrounding concrete to the rebar generates heat at the concrete–bar interface due to friction and mechanical interlocking. Additionally, the internal tensile stresses in the bar also heat up the rebar. This extra heat causes softening of the thermoset resin, which has low thermal resistance, on the bar surface. The friction-induced force transferring ability of the resin layer, which consists the outer surface of the rebar and is contact with the surrounding concrete, is reduced and the slip of the bar is facilitated by this heat. The radial thermal expansion coefficients of the aramid and carbon fibers is about three to five times higher than that of glass fibers. Hence, the increase in the radial pressure from this additional heat in AFRP and CFRP bars exceed the related increase in the GFRP bars [5,42]. Accordingly, the adherence to concrete, closely related to the radial pressure in the bar, is higher in AFRP and CFRP bars.

In the literature, a limited number of studies have been conducted on the effect of the modulus of elasticity of FRP on the concrete–FRP bond. Although various views on the effect of the change in the modulus of elasticity on adherence are presented in the literature [50,51,52,53,54], only a minority of these studies has the merit of directly focusing on the effect of the elastic modulus by fixing the other test parameters in the related experiments [25,55,56]. These studies unfolded that the sand-coated GFRP bars with high modulus of elasticity (HM) have lower bond strength values than the GFRP bars with low modulus of elasticity (LM). This unexpected outcome was attributed to the fact that the sand-coated surface layer was stripped from the rebar earlier in HM GFRP bars. This outcome invalidates the first of the two abovementioned explanations on the improved bonding properties of AFRP, BFRP and CFRP bars (the effect of shear lag) and even validates the reverse of this explanation. GFRP has the lowest elastic modulus among the four FRP types. The effect of elastic modulus on bond strength partially confirms the latter of the abovementioned explanations (the effect of heat on bond). The FRP bars with higher stiffness can absorb more energy when undergoing the same elongation as those with low axial stiffness. Hence, the bars with higher stiffness are expected to heat up more, causing the premature peeling of the sand-coating layer. This early peeling might exacerbate the concrete-FRP bond. In a related study in the literature, Arias et al. [57] reported that the FRP bars with matrix of higher strength exhibited improved bonding performance with concrete.

#### 3.1.3. Surface Texture (Surface Characteristic) of the Rebar

A good number of research studies in the literature are devoted to the influence of surface texture on concrete–FRP bar bond strength. Statistical, review and research studies on this topic are summarized in Table 3, which showcases an abundant number of surface types for FRP bars. Unlike the steel rebars, there exists no specifications or standards on the types and limitations of surface textures of FRP bars. The surface textures (finishes) of FRP bars are highly dependent on the preparation techniques and production process parameters of the manufactures due to the lack dimensional and material limitations in the related international and national standards. Identifying and comparing the effects of different surface types on the bond strength is cumbersome to achieve by also isolating this parameter from other test parameters. Even the same term can be observed to refer to completely different surface textures in different studies. Furthermore, there can be vast differences between the quality, grain size and density of the coating in sand-coated bar and the thickness, height and spacing of the ribs in the deformed and helically wound bars of different studies. For this reason, the comparison of the effects of reinforcement surface deformations on adherence cannot be put forward in clear terms. Nonetheless, a verbal and basic comparison was tried to be realized in the present review. In the discussions and comparison, the surface types were standardized (Figure 4 and Figure 5) according to the notations of Solyom and Balázs [58].

Accordingly,

R1 and R2 is the fine sand-coating (SCf).R3 and R4 shows the coarse sand-coating (SCr).R5 is the standard sand-coating (SC).R6 illustrates the helically wrapped (HW) surface.R7, R8 and R9 are the helically wrapped and sand-coated surface (HWSC).R10 shows the indented (In) surface.R11, R12 and R13 correspond to the ribbed (Rb) surface.

As shown in Table 3, the bond strength values of smooth bars remain well below the respective values of the bars with other surface types. In normal-strength concrete, the bond strength values of the bars with coarse sand-coating exceed the respective values of the fine sand-coated ones. In the presence of fine coating, the forces are transferred through only surface friction, while both mechanical interlocking and surface friction play role in the force transfer in the presence of coarse sand-coating. The change in the transfer mechanism can be held responsible for the improved bond behavior of the coarse sand-coated bars. As another important finding, the improvement of the bond behavior with the use of high-strength concrete in replacement for the normal-strength concrete is much more emphasized in the fine sand-coated bars when compared to the coarse sand-coated ones. High-strength concrete mixtures have better compaction and they contain less pores. The better compaction provides better penetration of concrete particles into the fine coating layer.

Hence, the adherence behavior alters into the mechanical interlocking with increasing concrete strength in bars with fine coating. Moreover, the surface areas of the fine sand-coated bars are larger due to the presence of indentations on the surface, and therefore, the contribution of improving the concrete quality to the bond strength becomes more considerable in these bars, having greater contact surface with concrete.

The other surface types, which have deeper surface deformations, convey the internal forces to the surrounding concrete through both friction and mechanical interlocking. The two-component transfer mechanism is the main reason for the higher bond strengths of these bars. The mechanical interlocking capacity changes with the rib height, rib spacing and rib thickness. From this point of view, with some exceptions, as the rib height increases and the rib spacing decreases, the bond strength increases. This increase stems from the increased surface area for the mechanical interlocking forces to develop. On the other side, significantly narrow and deep ribs might also lead to considerable reductions in the rigidity of ribs, and hence, lower limits for the spacing and upper limits for the rib height need to be established with the help of more detailed studies. In general, the bars with “In” surface type have larger rib spacings due to the increased rib thickness values and these bars possess lower bond strengths as compared to the bars with “Rb” surface texture. The lower bond strength values are caused by the fact that the force transfer in the indented bars rely mostly on the surface friction rather than mechanical interlocking. As in the case of indented bars, the bond strength values of the ribbed bars remain below the respective values of the helically wound bars.

In HW bars, the proportion of the forces transferred by the surface friction and mechanical interlocking varies with the height of the ribs. The contribution of mechanical interlocking increases with increasing rib depth, resulting in the increased adherence. Additionally, the bar starts to behave similar to a wedge with increasing rib height. Consequently, the shear forces are transmitted to the HW bars gradually, dissimilar to the sudden transfer of the shear forces in the FRP bars with In and Rb surface types. At around the peaks of the ribs, the mechanical interlocking forces turn completely into friction forces. This gradual transfer might also delay the shear failure of the beam by providing the transmission of shear forces along numerous interlaminar shear surfaces on the FRP bar instead of a single surface. The bars with “Rb” type of surface have superior bonding strength values when compared to the bars with “HW” type of surface.

### 3.2. Reinforcement Arrangement and Configuration

#### 3.2.1. Concrete Cover and Bar Spacing

The effects of concrete cover and bar spacing on FRP–concrete bond strength have also been subject to a variety of studies in the literature. The researchers have sought to identify the changes in the failure modes of FRP bars with changing bar spacing and concrete cover. Albeit an adequate number of studies were devoted to the effect of concrete cover, the bar spacing has caught the attention of only few researchers. Table 4 tabulates the statistical, review and research studies on the influence of concrete cover on the FRP–concrete bond.

The table clearly shows that the bond strength increases considerably with increasing concrete cover, which is thought to improve the confining pressure on the rebar. Besides this, the failure mode alters from concrete splitting to pull-out (bar debonding), and therefore, the confinement is improved with increasing concrete cover (Figure 6) [60,63]. The previous works depicted that the final failure is concrete splitting in the presence of a concrete cover of *d* (bar diameter) in all concrete types, i.e., NSC (normal-strength concrete), HSC (high-strength concrete), UHSC (ultra-high-strength concrete) [64,65]. When the concrete cover reaches 2*d*, the failure mode might turn into bar debonding or bar rupture depending on the embedment length [65]. In a related study, a clear cover of 2*d* (concrete cover of 2.5*d*) was reported to effectively prevent concrete splitting [35], while another study insists on a clear cover of 3*d* for avoiding the splitting failure completely [66]. ACI 440 1R-15 [5] states that the final failure will completely originate from bar debonding in the presence of a clear cover of 3*d* as long as the embedment length of the bar exceeds 19*d*.

The previous studies adopted a variety of surface textures and FRP mechanical properties. Furthermore, significant variations in concrete strength and embedment length makes it almost impossible to reach precise conclusions on the need for concrete cover for the complete prevention of concrete splitting. The concrete splitting failure is a result of the transfer of splitting forces to concrete [67], which is only possible when the rebar is subjected to excessive tensile stresses. Hence, the concrete splitting failure is probable only for certain clear cover and embedment length values [68]. As long as the concrete cover and embedment length exceed their respective threshold values, the rebar is expected to undergo debonding or tensile rupture failures [69]. That is why the critical values for the embedment length and concrete cover need to be specified to ascertain debonding or rupture failures and the critical values can only be identified with the help of additional experimental studies.

In summary, the FRP–concrete bond strength generally increases with increasing concrete cover since the confinement around the bar is improved with increasing cover thickness. The general tendency of the change in the FRP–concrete bond strength with increasing cover is complicated to specify as the failure mode also changes with increasing thickness of the concrete layer around the bar. Strictly speaking, the bond strength undergoes sudden changes while increasing the concrete cover, particularly at the transition of the failure mode from concrete splitting to debonding or rupture. Furthermore, concrete cover cannot be considered alone in the evaluation of test results, since this parameter governs the failure mode of a rebar together with the bar embedment length. Hence, the previous studies refrained from focusing on the simultaneous effects of concrete cover and embedment length on the FRP–concrete cover. Instead, only one of these two parameters changed in the related tests while keeping the other parameter fixed.

**Table 4 polymers-14-01796-t004:** Studies on the influence of concrete cover on FRP–concrete bond strength.

Ref.	Concrete Type and Concrete Strength	Embedment Length	Surface Type	Fiber Type and *d* (mm)	cc-First ^1^ (Failure Type)	cc-Second ^2^ (Failure Type)	Change in *τ* ^3^ (%)
[20]	UPC Cu ^4^ 98 MPa	4*d*	R (R)	Glass 10 mm	3*d* (P ^5^)	5*d* (P)	+10
[70]	NC 68 MPa	4*d*	R (In)	Glass 16 mm	1.5 (S ^6^), 2.50*d* (P)	2.50*d* (P), 6.25*d* (P)	+16, +19
[70]	NC 68 MPa	6*d*	R (In)	Glass 16 mm	1.50*d* (S), 2.50*d* (P)	2.50*d* (P), 6.25*d* (P)	+3, +15
[42]	SFRSCC ~64 MPa	5*d*, 20*d*	SC (SCf)	Glass 12, 12 mm	1.25*d* (P), 1.25*d* (P)	2.50*d* (P), 2.50*d* (P)	+21, +8
[42]	SFRSCC ~64 MPa	5*d*, 20*d*, 20*d*	R (In)	Glass 8, 8, 12 mm	1.88*d* (P), 1.88*d* (P), 1.25*d* (S)	3.75*d* (P), 3.75*d* (P), 2.50*d* (P)	−2, +17, +20
[35]	NC Cu 36 MPa	10*d*	Fine SC (SCf)	Glass 8 mm	2.00*d* (P)	3.00*d* (P)	+10
[35]	NC Cu 36 MPa	10*d*	Fine SC (SCf)	Carbon 8 mm	2.00*d* (P)	3.00*d* (P)	0
[35]	NC Cu 36 MPa	10*d*	R (HW or R)	Glass 8 mm	2.00*d* (P)	3.00*d* (P)	+15
[35]	NC Cu 36 MPa	10*d*	R (In)	Glass 8 mm	2.00*d* (P)	3.00*d* (P)	0
[71]	NC 50 MPa	5*d*	Fine SC (SCf)	Glass 16 mm	1.50*d* (S) 2.00*d* (P) 2.50*d* (P)	2.00*d* (P) 2.50*d* (P) 2.50*d* (P)	+27 −6 −15
[71]	NC 50 MPa	5*d*	Fine SC (SCf)	Glass 19 mm	1.50*d* (S−P) 2.00*d* (S) 2.50*d* (P)	2.00*d* (S) 2.50*d* (P) 2.50*d* (P)	+4 −6 −2
[72]	NC Cu 39 MPa	5*d*	R (In)	Glass 8 mm	1.25*d* (S) 2.50*d*	2.50*d* 8.88*d*	−11 −22
[72]	NC Cu 56 MPa	5*d*	R (In)	Glass 8 mm	1.25*d* (S) 2.50*d*	2.50*d* 8.88*d*	−6 −9
[37]	Soft computing techniques and statistical study						Increase
[49]	Artificial neural network and statistical study						Remarkable
[73]	Gauss process regression and statistical study						Increase
[74]	Review						Increase
[38]	Review						Increase

^1^ clear cover of reference bar; ^2^ clear cover of the compared bar; ^3^ the bond strength; ^4^ cubic; ^5^ pull-out failure; ^6^ concrete splitting.

**Figure 6 polymers-14-01796-f006:**
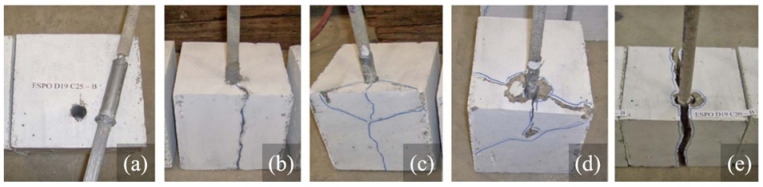
Bond failure types (**a**) pure pullout; (**b**) conventional concrete cover; (**c**) V-shaped concrete cover; (**d**) diagonal concrete cover; (**e**) concrete block split [71].

The number of studies on the influence of bar spacing on the FRP–concrete bond is rather limited and these studies showed that increasing the spacing between bars can contribute to the bond strength up to 50% [35,50]. If the spacing exceeds 7*d*, the bond strength was found to be unaffected by further increasing the bar spacing [35].

#### 3.2.2. Development (*l_d_*) or Embedment Length (*l_e_*)

There has been a great deal of research undertaken in the literature on the effects of embedment or development length on the FRP–concrete bond strength [22,24,38,47,70,75,76,77,78]. Two modes of behavior were reported in these studies. First, the non-uniform bond stress distribution with increasing development or embedment length reduces the maximum bond strength. The second observation is the reduced bond strength values due to the decrease in the friction, which is associated with the reductions in Poisson’s effect with increasing embedment length. Yet, higher adherence forces were stated to be conveyed with increasing embedment length as compared to the short lengths. The list of statistical, review and research studies on the effects of embedment or development length on bond strength are given in Table 5. With few exceptions, the tabulated results clearly imply that increasing the development length results in reductions in the bond strength value. However, the degree of this reduction is non-linear and dependent on the test parameters. (Figure 7) [18,20,24,78]. The adherence failure types are introduced in Section 3.2.1 with regard to the clear cover and embedment/development length.

#### 3.2.3. Reinforcement Position

The studies on the effects of reinforcement location concentrated on two positions, namely the lower and upper portions of the member, according to the direction of concrete cast. These studies generally concluded that the bond strengths of the upper bars are smaller than their lower counterparts since the water, air and fine aggregates in the mixture move upwards and accumulate underneath the rebars (Figure 8) [65,81,82].

In two of these studies [65,81], the bond strength values of the rebars were shown to drop up to 16% and 32% for an increase of 400 and 800 mm, respectively, in the concrete cast depth. In another study [82], the GFRP bar with a 150 mm greater cast depth was shown to have a 50% lower bond strength value than the bottom bar. Another research study reported a decrease of 74% in the bond strength with an additional concrete cast depth of 140 mm. Nonetheless, this reduction was stated to be due to possible bleeding and segregation in the concrete mixture. Additionally, a maximum decrease of 22% was reported in the same study for Self-Compacting Concrete (SCC) mixtures [83]. The statistical studies reported reduction in the bond strength and increase in the development length with increasing depth of the bar position in the member [37,74]. ACI 440.1R-15 [5] recommends a reduction of 33% for rebars with concrete cast depths exceeding 305 mm. Based on all these studies, the degree of influence of the concrete cast depth on the bond strength varies in a broad range. Yet, these reduction rates are affected by many additional factors, including but not limited to the bar diameter, bar surface texture, concrete mixture (W/C ratio, maximum aggregate size, grain size distribution), bar location, concrete cast and curing conditions. Moreover, for concrete cast depths above 200 mm and in the absence of any bleeding in concrete, the bond strength can be conservatively decreased by 50% and 20% in conventional concrete and SCC, respectively. The variation in the reduction in bond strength up to this depth can be assumed to be linear. Further and more detailed studies are needed on the subject.

#### 3.2.4. Confining Effect from Transverse Reinforcement

A great majority of the previous studies in the literature consisted of pull-out tests for determining the FRP–concrete bond strength, and hence, the number of studies on the effects of transverse reinforcement on the FRP–concrete bond is rather limited. The confining effect of transverse reinforcement on the longitudinal bars can only be reflected with the help of beam tests. There are two common opinions on this confining effect. The first one is the contribution of transverse reinforcement to bond strength through limitation of the crack widths in the member [49]. Accordingly, the bond strength can be increased up 29% with the help of the confining effect from the transverse reinforcement [84,85,86]. Yet, this effect also depends on the surface quality of the rebar [87]. The second view opposes the first one by implying that the hardness of the steel stirrups results in the peeling of bar surface during the bar slip and has an adverse effect on the bond strength [35]. Another statistical study reported the bond strength values to remain unaffected by the presence of transverse reinforcement [68]. According to these studies, the transverse reinforcement can be stated to have a positive or negative influence on the bar bond strength. Notwithstanding, there is a clear need for further studies on this very topic due to significant differences between the parameters and conditions of the related tests. In this respect, further studies are needed on the effects of transverse reinforcement on the FRP–concrete bond strength by also considering the concrete cover and bar surface texture as test variables.

### 3.3. Inherent Properties of Concrete

#### 3.3.1. Compressive Strength

The effect of concrete compressive strength on bond behavior may differ in steel and FRP reinforcing bars. Since steel rebars are homogeneous and isotropic as well as have a wholistic structure, bond failure patterns and stresses are governed by the shear strength of concrete [88]. However, owing to the composite structure of FRP, the bonding failure of FRP bars can be governed by the resin–fiber and resin–surface interlaminar shear stresses as well as the shear stresses in concrete (Figure 9) [21,88,89]. Table 6 lists the statistical, review and research studies on the effects of concrete compressive strength on the FRP–concrete bond. As can be seen in Table 6, the bond behavior of FRP rebars with different surface properties has been examined by previous researchers fora variety of concrete compressive strength values. The bond strength was found to increase with increasing concrete strength in almost all studies. The positive influence of increasing the concrete strength on bonding behavior arises from the restriction of internal crack propagation in concrete [72,90]. However, Table 6 also shows that the degree of influence of concrete strength on the FRP–concrete bond is variable in a wide range. There are also studies where the effect of increasing the concrete compressive strength on bonding behavior is limited due to the peeling of the outer surface of the FRP reinforcement from the resin or the interlaminar slip between the resin and fiber layers [21,74,89]. This limited effect depends on the fiber and resin densities, production types, fiber types, rigidities, maximum elongation rates and surface properties of the bars. If the shear strength of concrete is lower than the interlaminar shear strength, the failure originates from the shear failure of concrete, while the failure results from the stripping of the outer surface from the resin or interlaminar slip inside the matrix if the shear strength of concrete exceeds the internal shear strength of the FRP bar.

#### 3.3.2. Fiber Contribution

The effects of fiber addition to concrete mixture, which aims at controlling the crack widths by increasing the tensile strength of concrete, on the FRP–concrete bond has been subject to various studies in the literature. Some of these studies are given in Table 7. In these studies, a maximum fiber proportion of 1% by mass was used in the concrete mixture. The use of fibers tends to have a positive influence on FRP–concrete bond strength, while the degree of this influence can be rather diverse in different studies. This diversity is mostly due to the different surface textures of the bars in different studies although being assumed to originate from the type and amount of fibers in the concrete mixture. This surface texture affects the extent and distribution of initial cracks in concrete.

This cracking might also be affected by the fibers inside the mixture by improving the tensile strength of concrete and this effect will primarily depend on the density and length of fibers in the vicinity of the rebars (Figure 10). Nonetheless, significant variations were reported in certain studies despite the identical concrete compressive strength values, fiber type and densities and bar surface textures in these studies. These variations originate from two main reasons. First, concrete mixtures were not prepared and cast homogeneously in these studies. Secondly, the different maximum aggregate sizes and fiber lengths in these studies are thought to control the initiation and distribution of cracking in concrete and cause significant differences in bonding properties of the bars with various surface textures. Therefore, additional studies on fiber-added concrete mixtures with predefined and controlled maximum aggregate sizes and fiber lengths need to be conducted.

#### 3.3.3. Concrete Type

Various types of concrete were employed in the previous studies on the FRP–concrete bond (Table 8). However, the effect of concrete type on the FRP–concrete bond could not be made in the present review due to the scarcity of studies on the topic. Precise and accurate conclusions can only be achieved in the present of adequate studies.

## 4. Conclusions

The present paper is a detailed literature review on all parameters affecting the bond behavior of FRP reinforcing bars embedded in concrete. The influence of each parameter is discussed in the light of the findings of previous researchers. Precise and clear comments are given throughout the manuscript. The controversial and opposing comments of the previous researchers are not mentioned in the manuscript, since most of these comments originate from the differences between the testing conditions and test methods in different studies and negligence of certain parameters affecting the FRP–concrete bond. With the aim of not listing the inconsistent and ambiguous findings, only the following unquestionable conclusions are given in the present text together with the justifications behind each finding.

The bond strength of an FRP bar decreases with increasing bar diameter. This decrease is associated with three possible reasons. First, the slip of fiber layers within the resin, also known as the shear lag effect, is aggravated with increasing bar size and this effect has a negative impact on the FRP–concrete bond. Secondly, the amount of air voids and mixing water, accumulating underneath the bar, increases with increasing bar size and the weakness of the concrete around the bar results in the reduction of the bond strength. Finally, the mechanical interlocking and surface friction forces of a bar decrease as a result of the greater degree of Poisson’s effect on the bar with increasing bar diameter.

The bond strength values of GFRP bars are lower than the respective values of their CFRP, AFRP and BFRP counterparts, embedded in a similar concrete mixture. The lower adherence of GFRP to concrete stems from the more considerable shear lag effect in the GFRP bars due to the lower axial stiffness than the other three types of FRP. The greater slip of fibers from the core in GFRP results in the reduced bond strength values of these bars. Furthermore, the increase in the radial thermal expansion of the BFRP, AFRP and CFRP bars due to the friction at the bar–concrete interface improves the mechanical interlocking and surface friction of these bars in concrete, as compared to the GFRP bars, which are known to have smaller thermal expansion coefficient.

FRP bars with coarse sand-coating layer have higher bond strength values in normal-strength concrete than the bars with fine sand-coating layer. However, the bonding behavior of the fine sand-coated bars is improved to a greater extent with increasing concrete strength as compared to the coarse sand-coated bars. The better compaction and the lower amounts of air voids in high-strength concrete mixtures enable the fine particles of concrete to penetrate into the fine sand-coating layer and improve the bond behavior.

The mechanical interlocking mechanism is improved in ribbed bars with increasing rib height and decreasing rib spacing. The increase in the surface area for the development of mechanical interlocking forces results in the FRP–concrete bond strength to increase when using deeper ribs. However, further studies on the topic are needed to determine the minimum spacing and maximum height limits of the ribs since too closely-spaced and/or too deep ribs might reduce the rib rigidity and have adverse effects on the bond strength.

The thicker and more widely-spaced ribs in the bars with indented surface enables them to transmit greater surface friction forces as compared to the ribbed bars. Therefore, the bond strength values of the indented bars remain below the respective values of the ribbed and helically wrapped bars.

The concrete cast depth underneath an FRP bar influences the bond strength to a significant extent. With increasing cast depths, the amount of air voids and water underneath a bar increases, resulting in the compressive strength of concrete surrounding the bar and the FRP–concrete bond strength to decrease.

According to the existing studies in the literature, a clear concrete cover of at least three times the bar diameter is compulsory to avoid concrete splitting failure and to allow the debonding or tensile rupture failures to govern the specimen behavior. Increasing this spacing beyond seven times the bar diameter does not have a considerable effect on the FRP–concrete bond strength.

The contribution of increasing the compressive strength of concrete to FRP–concrete bond strength is bounded by upper limits. Increasing this strength contributes to the shear strength of the concrete layers around the bar, yet beyond certain limits of concrete strength, the peeling of the outer bar surface from the core and/or slip of the fibers inside the resin can trigger the bond failure of the bar rather than the shear failure of the surrounding concrete.

The bond strength tends to decrease with increasing embedment length of an FRP bar in concrete. The non-uniform stress distributions along the bar length and the reductions in the ability of a bar to convey the internal forces through surface friction are the primary reasons for the reduction in bond strength with increasing embedment length. This decrease follows a uniform path with increasing embedment length.

The transverse reinforcement definitely affects the FRP-concrete bond strength. However, further studies are needed to unfold the degree of this effect due to wide range of variation of the other test variables in the existing studies.

The maximum aggregate size and fiber length controls the initiation and spread of cracks in concrete based on the surface texture. Further studies are needed to uncover the effects of maximum aggregate size on the bond strengths of FRP bars embedded in the concrete mixtures with fibers.

The existing studies are not sufficient to specify the concrete cover boundaries for the change in the type of failure of FRP bars in concrete due to wide variations in the surface types and mechanical properties of the tested bars as well as the wide ranges of concrete strength and bar embedment length in the related tests. The concrete splitting failure necessitates the transfer of adequate splitting forces in concrete [61], which is only possible in the presence of specific clear cover and embedment length values [62]. Hence, detailed further studies related to the boundaries for the change of the failure mode from splitting to bar rupture or debonding are needed.

## Figures and Tables

**Figure 1 polymers-14-01796-f001:**
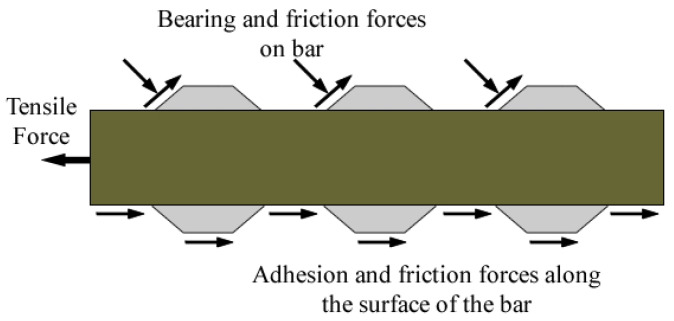
Bond mechanism of FRP bars (modified from [15]).

**Figure 2 polymers-14-01796-f002:**
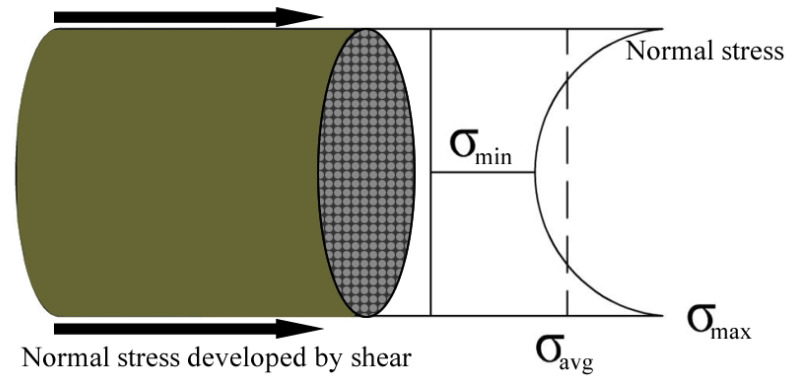
Shear lag effect (modified from [19,21]).

**Figure 3 polymers-14-01796-f003:**
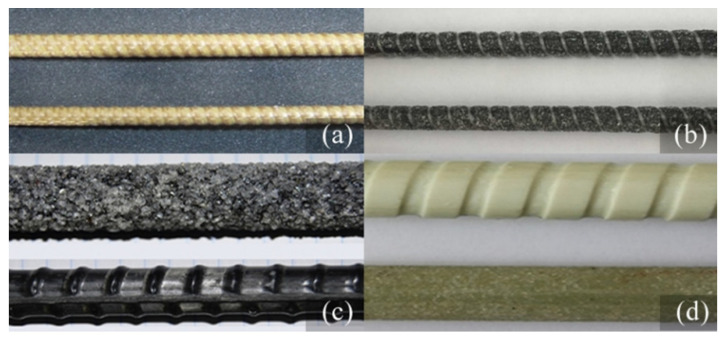
FRP bars and fiber types (**a**) Aramid [39]; (**b**) Basalt [40]; (**c**) Carbon [41]; (**d**) Glass [42].

**Figure 4 polymers-14-01796-f004:**
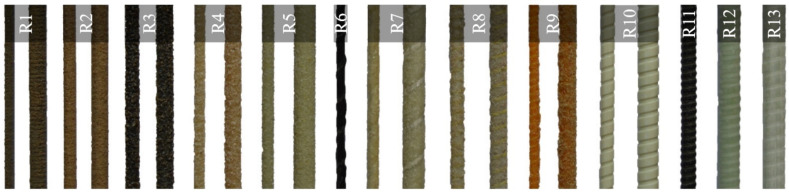
Surface deformation shapes of FRP rebars [58].

**Figure 5 polymers-14-01796-f005:**
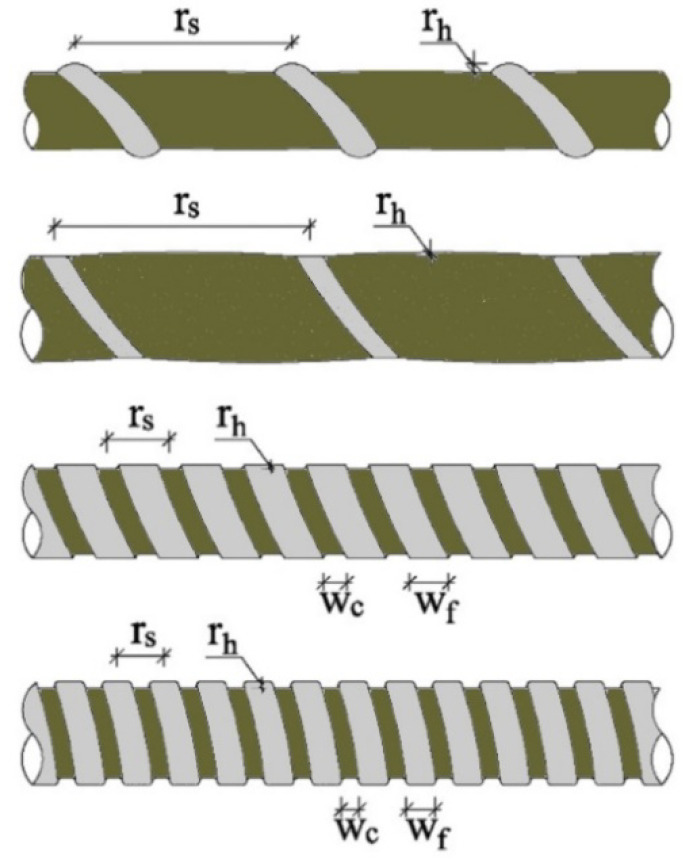
FRP bar rib height and spacing (modified from [36]).

**Figure 7 polymers-14-01796-f007:**
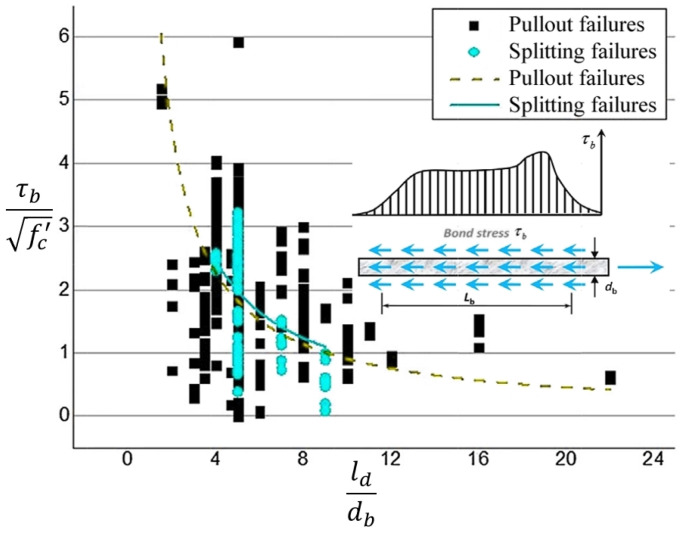
Effect of embedment length on bond strength [38].

**Figure 8 polymers-14-01796-f008:**
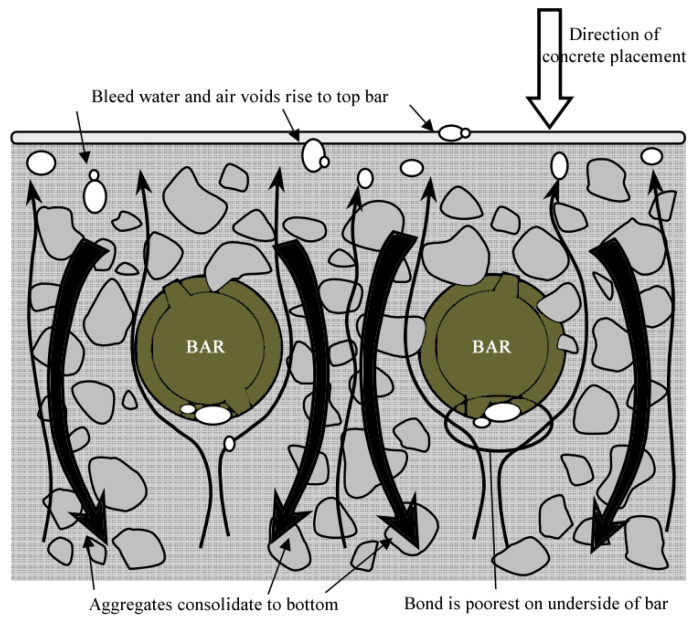
Effect of reinforcement position on bond strength [67].

**Figure 9 polymers-14-01796-f009:**
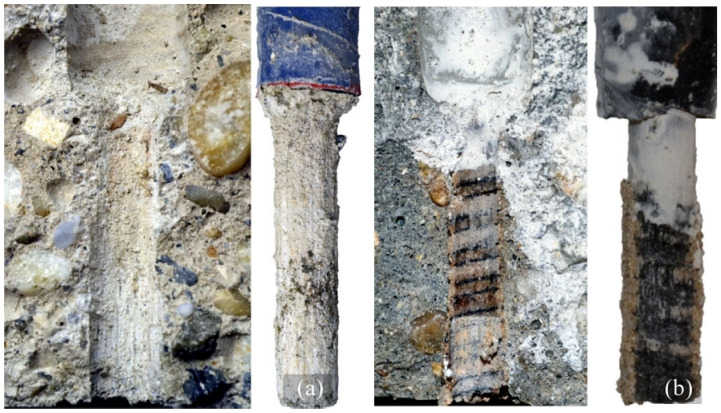
Failure types: (**a**) Concrete shear failure; (**b**) Peeling of rebar [36].

**Figure 10 polymers-14-01796-f010:**
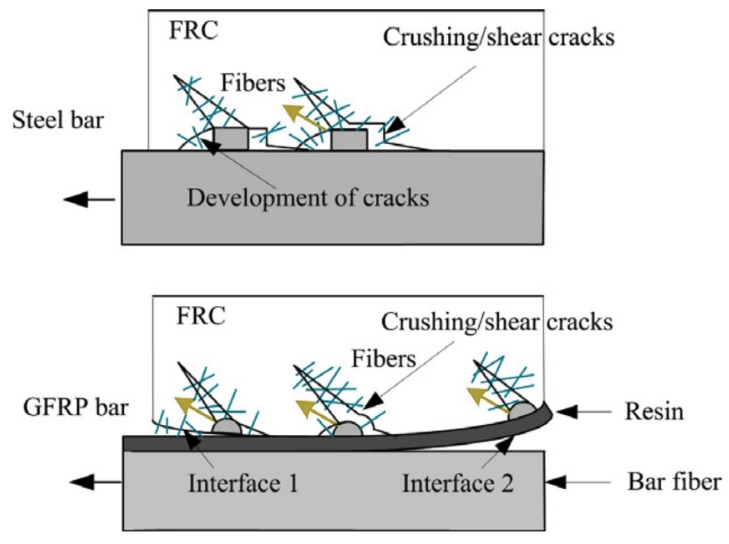
Effect of fibers on concrete cracks [97].

**Table 1 polymers-14-01796-t001:** Studies on the influence of bar diameter on FRP–concrete bond strength.

Ref.	Concrete Type and Strength	Embedment Length	Surface Type and Rib Dimensions	Fiber Type	*d_first_*^1^ (mm)	*d_sec_*^2^ (mm)	Change in Bond Stress (%)
[26]	NC ^3^ 35 MPa	5*d*	SC ^4^ (HW + SCf)	Glass	13	19	+15
[18]	SCC 54 MPa	80 mm	T (HW)	Basalt	12	20	−16
[27]	NC 35 MPa	5*d*	Thread In + SC (HW + SCf)	Glass	14	16	−59
[27]	NC 35 MPa	5*d*	Spirally wound (HW or R) rh ^5^ = 0.50 mm; rs ^6^ = 8.60 mm	Glass	14	16	−28
[28]	NC 28 MPa	6*d*	SC (SCc)	Glass	10	13	+29
[28]	NC 28 MPa	6*d*	B + SC (B + SCf)	Aramid	9	15	+38
[29]	CAC ~27 Mpa	5*d*	R (HW or R) rh = 0.26–0.28 mm; rs = 8.02–8.70 mm	Basalt	8	12	−6
[30]	CAC-S ~30 MPa	5*d*	Deep rib (HW or R) rh = 0.56–0.45 mm; rs = 11.00–10.00 mm	Carbon	8	12	−7
[30]	CAC-S ~30 MPa	5*d*	Deep rib (HW or R) rh = 0.56–0.60 mm; rs = 8.50–11.00 mm	Basalt	8	12	−8
[31]	RAC 35 MPa	5*d*	HW (HW or R)	Basalt	8 10	10 12	+29 −23
[32]	NC 33 MPa	5*d*	SC (SCf)	Basalt	8	12	−1
[33]	ECC Cu ^7^ 31 MPa	100 mm	R (HW or R) rh = 1.20 mm; rs = 9.4–10.2 mm	Glass	12 16	16 20	−11 −8
[34]	NC Cu 44 MPa	5*d*	HW (HW or R)	Basalt	10 14	14 20	−8 −24
[35]	NC Cu 36 MPa	10*d*	Fine SC (SCf)	Carbon	6 8	8 12	−3 −25
[35]	NC Cu 36 MPa	10*d*	Wound (In)	Glass	8	12	−11
[35]	NC Cu 36 MPa	10*d*	R (HW or R)	Glass	6 8	8 12	+15 −29
[36]	NC 35 MPa	5*d*	HW + SC (HW + SCm)	Glass	6 8	8 12	+35 −26
[36]	NC 35 MPa	5*d*	Coarse SC (SCc)	Basalt	6 8	8 12	+2 −22
[36]	NC 66 MPa	5*d*	Fine SC (SCf)	Basalt	6 8	8 12	+4 −24
[37]	Soft computing techniques and statistical						Decrease
[24]	Review						Not clear
[38]	Review						Decrease

^1^ the bar diameter in the first test; ^2^ the bar diameter in the subsequent test; ^3^ Concrete type notations are given in Section 3.3.3; ^4^ surface texture notations are given in Section 3.1.3; ^5^ rib height; ^6^ rib spacing; ^7^ cubic.

**Table 2 polymers-14-01796-t002:** Studies on the influence of fiber type on FRP–concrete bond strength.

Ref.	Concrete Type, Concrete Grade	Embedment Length	Surface Type, Rib Dimensions	Fiber Type	*d* (mm)	*τ_CF_*^2^/*τ*^1^	*τ*/*τ_Steel_*^3^
[26]	NC 35 MPa	5*d*	SC (HW + SCf)	Glass	13	−	0.78
[26]	NC 35 MPa	5*d*	SC (HW + SCf)	Glass	19	−	0.72
[18]	SCC 54 MPa	80 mm	T (HW)	Basalt	12	−	0.69
[18]	SCC 54 MPa	80 mm	HW + SC (HW + SCf)	Glass	12	−	0.48
[28]	NC 28 MPa	6*d*	S (S)	Glass	12	−	0.18
[28]	NC 28 MPa	6*d*	SC (SCc)	Glass	12	−	1.07
[28]	NC 28 MPa	6*d*	S (HW)	Carbon	10	−	0.60
[28]	NC 28 MPa	6*d*	SC (HW + SCf)	Carbon	10	−	0.70
[30]	CAC-S ~30 MPa	6*d*	Deep rib (HW or R) rh ^4^ = 0.56 mm; rs ^5^ = 11.00 mm	Carbon	8	Basalt − 1.08	1.35
[30]	CAC-S ~30 MPa	6*d*	Shallow rib (HW) rh = 0.20 mm; rs = 11.00 mm	Carbon	8	−	0.90
[30]	CAC-S ~30 MPa	6*d*	Deep rib (HW or R) rh = 0.56 mm; rs = 8.50 mm	Basalt	8	Carbon − 0.92	1.46
[30]	CAC-S ~30 MPa	6*d*	Deep rib (HW or R) rh = 0.45 mm; rs = 10.00 mm	Carbon	12	−	1.02
[30]	CAC-S ~30 MPa	6*d*	Deep rib (HW or R) rh = 0.60 mm; rs = 11.00 mm	Basalt	12	−	1.09
[32]	NC 33 MPa	5*d*	HW (HW)	Glass	10 10	Basalt − 1.88 Carbon − 2.83	−
[33]	ECC Cu ^6^ 31 MPa	100 mm	R (HW) rh = 0.20 mm; rs = 10.50 mm	Carbon	16	−	0.64
[43]	ECC Cu 31 MPa	100 mm	R (R) rh = 1.20 mm; rs = 9.70 mm	Glass	16	−	1.05
[44]	SCGC 40 MPa	5*d*	Spiral-wound (HW or R)	Basalt	10	−	0.72
[45]	NC 50 MPa	5*d*	Spiral ribs (HW or R) rh = 0.45 mm	Basalt	12	−	0.39
[45]	NC 50 MPa	5*d*	Spiral ribs + SC (HW + SCf) rh = 0.21 mm	Glass	13	−	0.51
[46]	NC ~40 MPa	20*d*	SC (SCc)	Glass	16	−	0.69
[35]	NC (Cu) 36 MPa	10*d*	Fine SC (SCf)	Glass	6 8 12	Carbon − 0.88 Carbon − 1.51 Carbon − 1.43	−
[47]	NC 35 MPa	5*d*	HW + SC (HW + SCf)	Aramid	6 8 10	Carbon − 1.69 Carbon − 1.34 Carbon − 1.16	− − 0.68
[48]	NC ~45 MPa	5*d*	R (R) rh = 0.71 mm	Glass	10	−	0.45
[49]	Artificial neural network and statistical						No effect
[25]	Review			Glass	−	Basalt − 1.29 Carbon − 0.92	−

^1^ the bond strength of test bar; ^2^ the bond strength of another comparable FRP bar; ^3^ the bond strength of the reference steel bar; ^4^ rib height; ^5^ rib spacing; ^6^ cubic.

**Table 3 polymers-14-01796-t003:** Studies on the influence of surface type on FRP–concrete bond strength.

Ref.	Concrete Type and Strength	Develo. Length	First Surface Type ^1^ and Ribs Dimensions	Second Surface Type ^2^ and Rib Dimensions	Fiber Type	*d* (mm)	Chan. in τ ^3^ (%)
[27]	NC ~30 MPa	5*d*	SC (HW + SCf)	Spirally wound (HW or R) rh ^4^ = 0.50 mm; rs ^5^ = 8.60 mm	Glass	14	+46
[27]	NC ~30 MPa	5*d*	SC (HW + SCf)	Deep thread In (HW or R) rh = 0.80 mm; rs = 10.00 mm	Glass	14	+32
[28]	NC 28 MPa	6*d*	S (S)	SC (SCc)	Glass	13	+506
[28]	NC 28 MPa	6*d*	S (HW)	SC (HW + SCf)	Carbon	10	+16
[30]	CAC-S ~30 MPa	5*d*	Shallow rib (HW) rh = 0.20 mm; rs = 11.00 mm	Deep rib (HW or R) rh = 0.56 mm; rs = 11.00 mm	Carbon	8	+51
[31]	RAC 35 MPa	5*d*	SC (SCf)	HW (HW or R) rs = 9.60 mm	Basalt	10	+96
[31]	RAC 35 MPa	5*d*	SC (SCf)	Screw thread (R) rs = 5.00 mm	Basalt	10	+78
[31]	RAC 35 MPa	5*d*	SC (SCf)	HW + SC (HW + SCf) rs = 14.00 mm	Basalt	10	+42
[32]	NC 33 MPa	5*d*	SC (SCf)	HW (HW or R)	Basalt	10	+70
[32]	NC 33 MPa	5*d*	SC (SCf)	Screw thread (R)	Basalt	10	+69
[32]	NC 33 MPa	5*d*	SC (SCf)	HW + SC (HW or R + SCf)	Basalt	10	+13
[59]	NC 30 MPa	5*d*	S (HW)	HW + SC (HW + SCf)	Glass	15	+7
[59]	NC 30 MPa	5*d*	S (HW)	HW (R)	Glass	15	+29
[60]	NC 40 MPa	5*d*	R (HW or R) rh = 0.50 mm; rs = 18.00 mm	R (HW or R) rh = 1.50 mm; rs = 18.00 mm	Glass	16	+55
[60]	NC 40 MPa	5*d*	R (HW or R) rh = 0.50 mm; rs = 18.00 mm	R (HW or R) rh = 0.50 mm; rs = 27.00 mm	Glass	16	+7
[36]	NC 35 MPa	5*d*	Fine SC (SCf)	Coarse SC (SCf)	Basalt	8 12	+21 +26
[36]	NC 35 MPa	5*d*	Fine SC (SCf)	Coarse SC (SCf)	Glass	6	+34
[36]	NC 35 MPa	5*d*	HW + SC (HW + SCm) rh = 0.53 mm; rs = 23.10 mm	R (R) rh = 0.46 mm; rs = 5.90 mm	Glass	12	+12
[36]	NC 35 MPa	5*d*	In (In) rh = 0.74 mm; rs = 8.74 mm	R (R) rh = 0.46 mm; rs = 5.90 mm	Glass	12	+24
[35]	NC Cu ^6^ 36 MPa	10*d*	Fine SC (SCf)	Wound (Grooves)	Glass	12	+66
[35]	NC Cu 36 MPa	10*d*	Fine SC (SCf)	R (HW or R)	Glass	12	+75
[57]	NC 23 MPa	5*d*	Fine SC (SCf)	Coarse SC (SCs)	Glass	9 16	+16 +9
[61]	NC 53 MPa	5*d*	Fine SC (SCf)	Grooved (In) rs = 9.00 mm	Glass	14	−4
[61]	NC 53 MPa	5*d*	Fine SC (SCf)	HW, SC (HW + SCf) rh = 0.47 mm; rs = 21.66 mm	Glass	14	+8
[61]	NC 57−47 MPa	5*d*	Grooved (In) rs = 9.00 mm	HW (HW) rh = 0.84 mm; rs = 16.13 mm	Glass	17	+32
[62]	NC 29 MPa	4*d*	R (R) rh = 0.60 mm; rs = 12.00 mm	R (R) rh = 0.60 mm; rs = 24.00 mm	Glass	12	−19
[62]	NC 29 MPa	4*d*	R (R) rh = 0.36 mm; rs = 12.00 mm	R (R) rh = 0.72 mm; rs = 12.00 mm	Glass	12	+61
[62]	NC 29 MPa	4*d*	R (R) rh = 0.60 mm; rs = 10.00 mm	R (R) rh = 0.60 mm; rs = 30.00 mm	Glass	10	−24
[42]	SFRSCC 61 MPa	5*d*	Fine SC (SCf)	Grooved (In) rh = 0.78 mm; rs = 8.50 mm	Glass	12–13	−12
[37]	Soft computing techniques and statistical study				Helical lugged > Spiral-wrapped > Sand-coated		
[49]	Artificial neural network and statistical study				No effect reported		
[24]	Review study				DS~ DS + SC > SC > S		

^1^ surface of the reference bar; ^2^ surface of the compared bar; ^3^ the bond strength; ^4^ rib height; ^5^ rib spacing; ^6^ cubic.

**Table 5 polymers-14-01796-t005:** Studies on the influence of reinforcement location on FRP–concrete bond strength.

Ref.	Concrete Type and Grade	Bar Diameter (mm)	Surface Type and Rib Dimensions	Fiber Type	*l_e-first_*^1^ (mm or d) (Failure)	*l_e-second_*^2^ (mm or d) (Failure)	Change in τ ^3^ (%)
[18]	SCC 54 MPa	12	T (HW)	Basalt	40 (P ^4^) 80 (P)	80 (P) 120 (P)	−27 −40
[29]	CAC ~27 MPa	10	R (HW or R) rh ^5^ = 0.36 mm; rs ^6^ = 9.02 mm	Basalt	5*d* (P) 10*d* (P)	10*d* (P) 12*d* (R ^7^)	−9 −4
[30]	CAC-S ~30 MPa	8	Deep rib (HW or R) rh = 0.56 mm; rs = 11.00 mm	Carbon	5*d* (P)	7.5*d* (P)	−1
[30]	CAC-S ~30 MPa	8	Shallow rib (HW) rh = 0.20 mm; rs = 11.00 mm	Carbon	5*d* (P)	7.5*d* (P)	−13
[30]	CAC-S ~30 MPa	12	Deep rib (HW or R) rh = 0.60 mm; rs = 11.00 mm	Basalt	5*d* (P)	7.5*d* (S)	−16
[44]	SCGC 40 MPa	10	Spiral-wound (HW or R)	Basalt	5*d* (P) 10*d* (P)	10*d* (P) 15*d* (P)	−31 −21
[79]	GPC 42 MPa	8	R (R)	Basalt	5*d* (P) 10*d* (R)	10*d* (R) 15*d* (P)	−2 −54
[80]	NC Cu ^8^ 38 MPa	8 12	(HW or R)	Basalt	10*d* (P) 10*d* (P)	20*d* (R) 20*d* (R)	−38 −30
[57]	NC 23 MPa	16	Fine SC (SCf)	Glass	5*d* (P)	10*d* (P)	−48
[57]	NC 23 MPa	16	Coarse SC (SCc)	Glass	5*d* (P)	10*d* (P)	−46
[37]	Soft computing techniques and statistical study					Nonlinear Decrease	
[49]	Artificial neural network and statistical study					Decrease	
[68]	Statistical study					Decrease	
[24]	Review study					Decrease	
[38]	Review study					Decrease	

^1^ embedment length of reference bar; ^2^ embedment of the compared bar; ^3^ the bond strength; ^4^ debonding (pull-out) failure; ^5^ rib height; ^6^ rib spacing; ^7^ tension rupture; ^8^ cubic.

**Table 6 polymers-14-01796-t006:** Studies on the influence of concrete compressive strength on FRP–concrete bond strength.

Ref.	Concrete Typeand Specimen Shape	Devel.Length	Surface Type	Fiber Type and Bar Diameter (mm)	*f_c-first_*^1^ (MPa)	*f_c-second_*^2^ (MPa)	Change in τ ^3^ (%)
[72]	RAC Cube	5*d*	Sand Coated (SCf)	Carbon 12	34 47	47 63	+3 +46
[72]	RAC Cube	5*d*	SC with shallow spiral In (HW + SCc)	Basalt 12	34 47	47 63	+12 +1
[72]	RAC Cube	5*d*	R (HW or R)	Glass 12	34 47	47 63	−9 +13
[29]	CAC Cube	10*d*	R (HW or R) rh ^4^ = 0.36 mm; rs ^5^ = 9.02 mm	Basalt 10	16 22	22 32	+21 +57
[60]	NC Cube	5*d*	R (HW or R) rh = 0.50 mm; rs = 18.00 mm	Glass 12	25 49	49 64	+100 +15
[33]	ECC Cube	100 mm	R (R) rh = 0.20 mm; rs = 10.5 mm	Carbon 16	31	70	+163
[57]	NC −	5*d*	Fine SC (SCf)	Glass 9 and 16	23	56	+18 +2
[57]	NC −	5*d*	Coarse SC (SCc)	Glass 9 and 16	23	56	+34 +16
[75]	NC −	5*d*	(R)	Basalt 10	37 55	55 73	+109 +23
[91]	NC −	285 mm 380 mm	HW + SC (HW + SCf)	Glass 19	32 31	42 39	+13 +16
[61]	NC −	5*d*	Grooved (In)	Glass 8 and 16	30 27	53 57	+36 +44
[61]	NC −	5*d*	HW, SC (HW + SCf)	Glass 13	29	51	+73
[36]	NC −	5*d*	Fine SC (SCf)	Glass 8 and 12	35 35	66 66	+90 +94
[36]	NC −	5*d*	Coarse SC (SCf)	Glass 8 and 12	35 35	66 66	+36 +64
[36]	NC −	5*d*	In (In) rh = 0.74 mm; rs = 8.74 mm	Glass 8 and 12	35 35	66 66	+65 +50
[36]	NC −	5*d*	R (R) rh = 0.46 mm; rs = 5.90 mm	Glass 8 and 12	35 35	66 66	+87
[34]	NC cube	5*d*	HW (HW or R)	Basalt 10	44	72	+19
[63]	NC −	5*d*	SC (SCf)	Glass 16and 19	71 129 148 71 129 148	129 148 175 129 148 175	−1 +7 −3 +15 −4 −1
[88]	NC −	4*d*	SC (SCf)	Glass 13	26 41	41 92	+7 +18
[88]	NC −	4*d*	HW	Glass 13	26 41	41 92	+14 +21
[88]	NC −	4*d*	(HW)	Steel 13	26 41	41 92	+13 +47
[37]	Soft computing techniques and statistical study					Linear Decrease	
[74]	Review study					Increase	

^1^ concrete strength of reference bar; ^2^ concrete strength of the compared bar; ^3^ bond strength; ^4^ rib height; ^5^ rib spacing.

**Table 7 polymers-14-01796-t007:** Studies on the influence of fiber contribution on FRP–concrete bond strength.

Ref.	Concrete Fiber Type and Length (mm)	Develop. Length (mm)	Surface Type and Rib Dimensions	Fiber Type and Bar Diameter (mm)	Original Concrete Strength ^1^ (MPa)	Fiber Concrete Strength ^2^ (MPa) &Fiber Vol.	Change in τ ^3^ (%)
[92]	Aramid (30–40)	150	SC (SCf)	Glass 12	Cu ^4^ 47	Cu 43 (%0.5)	~0
[92]	Aramid (30–40)	150	Helically deformed (HW or R)	Glass 12	Cu 47	Cu 43 (%0.5)	~0
[92]	Aramid (30–40)	150	R (HW or R)	Glass 12	Cu 47	Cu 43 (%0.5)	~+20
[93]	Polyolefin structural synthetic(30)	5*d*	(HW)	Carbon 9	74	75 (%1)	+25
[93]	PVA (30)	5*d*	(HW)	Carbon 9	74	75 (%1)	+30
[94]	Steel (30)	5*d*	SC (SCf)	Glass 13	59 59	57 (%0.5) 58 (%1.0)	+53 +10
[94]	Polypropylene (30)	5*d*	SC (SCf)	Glass 13	59 59	52 (%0.5) 56 (%1.0)	+29 +36
[94]	PVA (30)	5*d*	SC (SCf)	Glass 13	59 59	54 (%0.5) 63 (%1.0)	+45 +54
[94]	Steel (30)	5*d*	HW (HW)	Glass 13	59 59	57 (%0.5) 58 (%1.0)	+3 +8
[94]	Polypropylene (30)	5*d*	HW (HW)	Glass 13	59 59	52 (%0.5) 56 (%1.0)	−5 −4
[94]	PVA (30)	5*d*	HW (HW)	Glass 13	59 59	54 (%0.5) 63 (%1.0)	0 −5
[95]	Steel (32)	4*d*	Two directional rib (HW or R)	Carbon 9	94 94	96 (%0.25) 103 (%0.5)	+24 +57
[95]	Steel (32)	4*d*	One directional rib (HW or R)	Carbon 9	94 94	96 (%0.25) 103 (%0.5)	+33 +67
[18]	Polypropylene (12)	80 mm	T (HW)	Basalt 12	54 54	50 (%0.30) 49 (%0.60)	−16 −24
[18]	Polypropylene (12)	80 mm	HW + SC (HW + SCf)	Glass 12	54 54	50 (%0.30) 49 (%0.60)	−39 −61
[96]	Glass (18) (36) (50)		HW + SC (HW + SCf)	Glass 10	49 49 49 49	50 (%0.50) 54 (%1.00) 50 (%0.50) 54 (%1.00)	−5 −4 −3 0
[96]	Glass (18) (36) (50 mm)		R (R) rh ^5^ = 0.80 mm rs ^6^ = 10.50 mm	Basalt 10	49 49 49 49	50 (%0.50) 54 (%1.00) 50 (%0.50) 54 (%1.00)	−13 −11 −5 −5

^1^ plain mixture; ^2^ fiber added mixture; ^3^ bond strength; ^4^ cubic; ^5^ rib height; ^6^ rib spacing.

**Table 8 polymers-14-01796-t008:** Studies on the influence of concrete type on FRP–concrete bond strength.

Ref.	Specimen Shape and Embedment Length	Surface Type	Fiber Type And Bar Diameter (mm)	First Concrete Type and Strength (MPa)	Second Concrete Type and Strength (MPa)	Change in τ ^1^ (%)
[46]	− 40*d*	SC (SCc)	Glass 16	NC 39	SCC 41	+5
[46]	− 40*d*	SC (SCc)	Glass 16	NC 39	SCC 41	−18
[98]	Cu ^2^ 3*d*	R (R)	Glass 16	NC 48	SCC 45	+13
[98]	Cu 3*d*	R (R)	Glass 16	NC 65	SCC 60	+9
[72]	Cu 5*d*	SC (SCf)	Carbon 12	NC 37	RAC 34	−2
[72]	Cu 5*d*	SC with shallow spiral In (HW + SCc)	Basalt 12	NC 37	RAC 34	+3
[72]	Cu 5*d*	R (HW or R)	Glass 12	NC 37	RAC 34	+24
[26]	− 5*d*	SC (HW + SCf)	Glass 13 19	NC 37	HVFAC (%50 rep.) 30	−18 −7
[26]	− 5*d*	SC (HW + SCf)	Glass 13 19	NC 37	HVFAC (%50 rep.) 30	−29 −28
[99]	− 2*d*	R (R)	Basalt 10	HPC 82	UHPC 137	~0
[99]	− 2*d*	R (R)	Glass 10	HPC 82	UHPC 137	~0
[99]	− 2*d*	Deformed	Steel 10	HPC 82	UHPC 137	+38
[45]	Cu 65 mm	Spiral ribs + SC (HW + SCf) rh ^3^ = 0.21 mm	Glass 13	MPC 48	MPC-S 49	+14
[45]	Cu 65 mm	Spiral ribs + SC (HW + SCf) rh = 0.21 mm	Glass 13	NC 50	NC-S 48	−13
[45]	Cu 65 mm	Spiral ribs (HW or R) rh = 0.45 mm	Basalt 12	NC 50	MPC 48	+51
[45]	Cu 65 mm	Spiral ribs + SC (HW + SCf) rh = 0.21 mm	Glass 13	NC 50	MPC 48	+24
[45]	Cu 65 mm	R (R)	Steel 13	NC 50	MPC 48	−1
[96]	− 50 mm	HW + SC (HW + SCf)	Glass 10	SSSC (%0 EA) 49	SSSC (%6 EA) 54	+10
[96]	− 50 mm	R (R) rh = 0.80 mm; rs ^4^ = 10.50 mm	Basalt 11	SSSC (%0 EA) 49	SSSC (%6 EA) 54	−13
[31]	− 5*d*	HW (HW or R)	Basalt 8	NC 43	RAC 35	−31
[31]	− 5*d*	HW (HW or R)	Basalt 8	NC 43	SSC 45	+1
[31]	− 5*d*	HW (HW or R)	Basalt 8	SSC 45	SSC + RAC 40	−19
[100]	Cu 90 mm	R (HW or Ribbed)	Basalt 14	NC 51	SSC-S 47	+41
[100]	Cu 90 mm	SC (HW or R +SCm)	Basalt 16	NC 51	SSC-S 47	+8

^1^ bond strength; ^2^ cubic specimen; ^3^ rib height; ^4^ rib spacing.

## Data Availability

The data, presented within the present paper, is available upon request.

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
