# Peer review of "A Review on the Physical Parameters Affecting the Bond Behavior of FRP Bars Embedded in Concrete"

_polymers, 2022, doi:10.3390/polym14091796_

Round 1

Reviewer 1 Report

The paper focuses on the review on the physical parameters affecting the bond behavior of FRP bars embedded in concrete. Even though lots of research work has been summarized to analyze the influence of physical parameters on interfacial bond strength, the mathematical curves to describe qualitatively and quantitatively the effects of these parameters is missing. From some current tables, it is difficult to analyze the influence of these parameters. Therefore, it is suggested that the authors refer to the following specific comments to greatly improve the quality of the paper and provide a clear reviewing thought for the readers.

  1. In the abstract part, the main purpose of the current research should be further clarified. In addition, the background analysis is also essential.
  2. In the first paragraph of the introduction, the authors analyzed the advantages and disadvantages of FRP bars. No distinction between the types of FRP brought about some imprecise conclusions. The advantages of FRP performance are dependent on the fiber type. For CFRP, it has high tensile strength, elastic modulus, low elongation at break, excellent corrosion, fatigue and creep resistances. However, its high price and electrochemical corrosion when contacting with metal materials in a humid environment may be the main problems. For GFRP, the tensile strength and elastic modulus are relatively low. In contrast, its elongation at break is higher than CFRP. However, its long-term performance (mechanical, fatigue and creep performances) under the coupling of environment (temperature, humidity and corrosive medium) and loading is poor. Therefore, it is suggested that the authors should analyze the advantages and disadvantages of FRP based on the fiber type and further clarify the engineering application of different FRPs. Please see the latest research work on CFRP and GFRP below.

Creep performance: Composite Structures, 2022. 281: 115060.

Fatigue performance: Materials Today: Proceedings, 2021, 46: 555-561.

Corrosive performance: Materials Research and Technology, 2021, 14:2812-2831.

  1. Line 40-41, the low transverse and lateral strength values as compared to the longitudinal tensile strength led to a problem of successful anchorage of FRP bars. It is suggested that the authors refer to relevant work (such as: anchorage system of FRP bar or rod) for further expansion and analysis.
  2. Ten main parameters affecting the FRP-concrete bond are listed under four headings, including the inherent properties of FRP rebars; the arrangement and configuration of reinforcement; the inherent properties of concrete and the method of testing. However, the above four main factors may only affect the original interface bonding performance of FRP-concrete without the aging. When FRP bar reinforce concrete structure is used in the actual engineering, the service environment (temperature and humidity, corrosive medium type, static/dynamic loading and service time, etc.) may be more important compared to the above parameters, leading to the degradation of FRP-concrete interface bonding performance. The authors are suggested to provide relevant analysis and explanation based on the summary of durability.
  3. When analyzing the influence of the diameter of FRP bar on the interfacial bonding properties, the mathematical relationship curve between bar diameter and bonding strength should be further established according to the data from Table 1. The current writing is difficult to distinguish the effect of bar diameter on interfacial bond strength. Similar situations also apply to the effects of other parameters.
  4. When analyzing the influence of fiber type on FRP-concrete bond strength, the surface type and rib dimensions may be two important factors of interface bond strength. However, the above two factors have no essential relationship with the type of fiber. They are more likely to depend on the preparation method and production process parameters. Please provide relevant explanations.
  5. In part 3.1, the three influencing factors should have coupled effect on the interface bond strength of FRP-concrete. Furthermore, the variations of interfacial bonding strength with diameter, fiber type, elastic modulus and surface texture should be comprehensively analyzed through an orthogonal experimental data analysis.
  6. What is the influence of concrete cover on FRP-concrete bond strength? From Table 4, it is difficult to see the change trend due to less data and more variable parameters.
  7. Please add the qualitative and quantitative analysis and conclusions on the effect of embedding length and confining effect from transverse reinforcement on the bonding strength in part 3.2.2 and 3.2.4.
  8. In part 3.3, the relationship among concrete compressive strength, fiber contribution, concrete type and interface bonding strength should be further established through the description of mathematical curve.
  9. The current conclusion is too long and need further refinement, only including the key information.

Reviewer 2 Report

Title: A Review on the Physical Parameters Affecting the Bond Behavior of FRP Bars Embedded in Concrete

The referee would like to recommend this work to major revision and to be published after consideration according to the comments below:

  1. The writing of this paper needs to be improved and polished. Some clumsy and neglectful expressions can be found. Some of them are highlighted in the article.

  1. Please add more current papers in the literature and improve introduction section. Some interesting papers related to the topic of this manuscript could be:

Theoretical study of stress transfer in platelet reinforced composites. Journal of Theoretical and Applied Mechanics, (2014) 52(1), 3-14.

Thermo-mechanical stress analysis in platelet reinforced composites with bonded and debonded platelet end. Journal of Mechanical Science and Technology, (2015) 29(5), 2067-2072.

Calcium carbonate nanoparticles effects on cement plast properties. Microsystem Technologies, (2021)  3059–3076.

  1. Some Figures like as Figure. 6 are unclear. Please correct them.

Round 2

Reviewer 1 Report

The authors have responded well to the comments of the reviewer. It is suggested to accept the current paper.

Reviewer 2 Report

Accepted.